# Protective Effect of Rutin on Triethylene Glycol Dimethacrylate-Induced Toxicity through the Inhibition of Caspase Activation and Reactive Oxygen Species Generation in Macrophages

**DOI:** 10.3390/ijms231911773

**Published:** 2022-10-04

**Authors:** Li-Chiu Yang, Yu-Chao Chang, Kun-Lin Yeh, Fu-Mei Huang, Ni-Yu Su, Yu-Hsiang Kuan

**Affiliations:** 1Department of Dentistry, Chung Shan Medical University Hospital, Taichung 40201, Taiwan; 2School of Dentistry, Chung Shan Medical University, Taichung 40201, Taiwan; 3Department of Veterinary Medicine, National Chung Hsing University, Taichung 40227, Taiwan; 4Department of Pharmacology, School of Medicine, Chung Shan Medical University, Taichung 40201, Taiwan; 5Department of Pharmacy, Chung Shan Medical University Hospital, Taichung 40201, Taiwan

**Keywords:** TEGDMA, rutin, macrophage, cytotoxicity, genotoxicity, apoptosis, reactive oxygen species, antioxidant system

## Abstract

Rutin, also called quercetin-3-rhamnosyl glucoside, is a natural flavonol glycoside present in many plants. Rutin is used to treat various diseases, such as inflammation, diabetes, and cancer. For polymeric biomaterials, triethylene glycol dimethacrylate (TEGDMA) is the most commonly used monomer and serves as a restorative resin, a dentin bonding agent and sealant, and a bone cement component. Overall, TEGDMA induces various toxic effects in macrophages, including cytotoxicity, apoptosis, and genotoxicity. The aim of this study was to investigate the protective mechanism of rutin in alleviating TEGDMA-induced toxicity in RAW264.7 macrophages. After treatment with rutin, we assessed the cell viability and apoptosis of TEGDMA-induced RAW264.7 macrophages using an methylthiazol tetrazolium (MTT) assay and Annexin V-FITC/propidium iodide assay, respectively. Subsequently, we assessed the level of genotoxicity using comet and micronucleus assays, assessed the cysteinyla aspartate specific proteinases (caspases) and antioxidant enzyme (AOE) activity using commercial kits, and evaluated the generation of reactive oxygen species (ROS) using a dichlorodihydrofluorescein diacetate (DCFH-DA) assay. We evaluated the expression of heme oxygenase (HO)-1, the expression of nuclear factor erythroid 2 related factor (Nrf-2), and phosphorylation of AMP activated protein kinase (AMPK) using the Western blot assay. The results indicated that rutin substantially reduced the level of cytotoxicity, apoptosis, and genotoxicity of TEGDMA-induced RAW264.7 macrophages. Rutin also blocked the activity of caspase-3, caspase-8, and caspase-9 in TEGDMA-stimulated RAW264.7 macrophages. In addition, it decreased TEGDMA-induced ROS generation and AOE deactivation in macrophages. Finally, we found that TEGDMA-inhibited slightly the HO-1 expression, Nrf-2 expression, and AMPK phosphorylation would be revered by rutin. In addition, the HO-1 expression, Nrf-2 expression, and AMPK phosphorylation was enhanced by rutin. These findings indicate that rutin suppresses TEGDMA-induced caspase-mediated toxic effects through ROS generation and antioxidative system deactivation through the Nrf-2/AMPK pathway. Therefore, rutin has the potential to serve as a novel antitoxicity agent for TEGDMA in RAW264.7 macrophages.

## 1. Introduction

Rutin, also called quercetin-3-rhamnosyl glucoside, is a natural flavanol glycoside present in the aglycone quercetin and the disaccharide rutinose. Rutin is a bioflavonoid antioxidant and is found as an abundant component in various plants, such as grapes, buckwheat, tea, apples, tobacco, *Forsythia*, fruits, vegetables, and grains [1,2]. Rutin has various nutritional benefits and medicinal applications, for example, in inflammation, oxidative stress, diabetes, hyperglycemia, cancer, and genotoxicity [3,4,5,6]. Pretreatment with rutin reduced inflammatory responses, genotoxicity, and lung toxicity in mice after exposure to benzo[*a*]pyrene [B(α)P], an environmental pollutant and a potentially carcinogenic substance [7]. In the innate immune system, both rutin and related extracts from natural plants reduce the levels of cytotoxicity and genotoxicity of macrophages through various chemical substances, such as bisphenol A-glycidyl methacrylate (BisGMA) and lipopolysaccharide (LPS) [8,9]. Therefore, to determine how rutin can serve as a novel protective agent against immunotoxic diseases induced by various biomaterials, understanding how rutin reduces the levels of cytotoxicity and genotoxicity of macrophages is crucial.

Polymeric biomaterials have been widely used in the fields of odontology and orthopedics as restorative resins, dentin bonding agents and sealants, and bone cement components. For polymeric biomaterials, triethylene glycol dimethacrylate (TEGDMA) is the most commonly used monomer [10,11]. After polymerization, the residual monomer released from TEGDMA-based polymeric biomaterials causes injury to peripheral tissues or cells [12]. Macrophage is a type of tissue-resident phagocyte and plays an important role in the first line of defense against invasive pathogens and in the destruction of apoptotic cells [13]. Production of reactive oxygen species (ROS) participates in the antimicrobial, antiparasitic, antivirus, and immunoregulatory functions in macrophages. There are several sources of ROS generation, including cytosolic NADPH oxidase, cytosolic xanthine oxidase, and mitochondrial electron transport chain [14,15]. However, excess ROS production can cause peripheral tissue and cellular damage, DNA damage, lipid peroxidation, and pro-inflammatory response [14,15,16]. Recent several studies have shown that cytotoxicity was induced by TEGDMA in RAW264.7 macrophages via their large interaction potency and impregnation into lipid bilayers [17]. The incubation of macrophages with TEGDMA leads to various proinflammatory responses, such as the upregulation of cyclooxygenase-2 (COX-2) and inducible nitric oxide synthase (iNOS) [17,18]. TEGDMA induces cytotoxicity via apoptosis and genotoxicity due to DNA damage and cysteinyl aspartate-specific proteinase (caspase) activation in macrophages [19]. Furthermore, TEGDMA induces apoptosis through generation of ROS, phosphorylation of mitogen-activated protein kinase (MAPK) and downstream transcription factor [20,21]. In recent studies, rutin effectively prevented BisGMA-induced toxicity in macrophages and UV-induced dysfunction in skin fibroblasts by down-regulation of ROS generation and up-regulation of AOE activity and expression [8,22,23]. In the present study, we primarily investigated how rutin protected against the toxicity of TEGDMA via downregulation of ROS generation and the relative molecular mechanism in macrophages.

## 2. Results

### 2.1. Effects of Rutin on Cytotoxicity Induced by Triethylene Glycol Dimethacrylate (TEGDMA)

As shown in Figure 1, treatment with 3 μM TEGDMA significantly induced cytotoxicity in RAW264.7 cells compared with the control group (*p* < 0.05). However, pretreatment with rutin reduced TEGDMA-induced cytotoxicity in a concentration-dependent manner, with the reduction becoming significant at 30 μM (*p* < 0.05).

### 2.2. Effects of Rutin on Necrosis and Apoptosis Induced by Triethylene Glycol Dimethacrylate (TEGDMA)

An Annexin V-FITC/propidium iodide apoptosis detection kit was used to define the mode of cell death, including necrosis and apoptosis, induced by TEGDMA in RAW264.7 cells (Figure 2). After treating cells with 3 μM TEGDMA, necrotic and apoptotic populations significantly increased compared with the control group (*p* < 0.05). Pretreatment with rutin, however, decreased TEGDMA-induced apoptosis and necrosis in a concentration-dependent manner, with the decrease becoming significant at 30 μM (*p* < 0.05).

### 2.3. Effects of Rutin on Genotoxicity Induced by Triethylene Glycol Dimethacrylate (TEGDMA)

Comet and MN assays were used to study the protective effect of rutin on TEGDMA-induced DNA damage. As shown in Figure 3, treatment with 3 μM TEGDMA significantly induced DNA damage, including the tail moment, tail length, and MN formation of RAW264.7 cells (*p* < 0.05). Pretreatment with rutin, however, reduced TEGDMA-induced genotoxicity in a concentration-dependent manner, with the reduction becoming significant at 30 μM (*p* < 0.05).

### 2.4. Effects of Rutin on Caspase Activity Induced by Triethylene Glycol Dimethacrylate (TEGDMA)

Treatment with 3 μM TEGDMA significantly induced caspase-3, caspase-8, and caspase-9 activity (*p* < 0.05). However, pretreatment with rutin reduced TEGDMA-induced caspase-3, caspase-8, and caspase-9 activity in a concentration-dependent manner, with the reduction becoming significant at 30 μM (*p* < 0.05, Figure 4).

### 2.5. Effects of Rutin on Reactive Oxygen Species Generation Induced by Triethylene Glycol Dimethacrylate (TEGDMA)

As shown in Figure 5, 3 μM TEGDMA significantly induced ROS generation (*p* < 0.05). However, pretreatment with rutin reduced TEGDMA-induced ROS generation in a concentration-dependent manner, with the reduction becoming significant at 30 μM (*p* < 0.05).

### 2.6. Effects of Rutin on Antioxidant Enzyme Activity by Triethylene Glycol Dimethacrylate

As shown in Figure 6, 3 μM TEGDMA significantly reduced AOE activity, including that of SOD and CAT (*p* < 0.05). However, pretreatment with rutin revised TEGDMA-reduced AOE activity. In addition, pretreatment with rutin enhanced the AOE activity on TEGDMA-treated RAW264.7 macrophages in a concentration-dependent manner, with the upregulation becoming significant at 30 μM (*p* < 0.05).

### 2.7. Effects of Rutin on HO-1 and Nrf-2 Expression on Triethylene Glycol Dimethacrylate (TEGDMA)-Treated RAW264.7 Macrophages

As shown in Figure 7, expression of Nrf-2 and HO-1 was reduced by TEGDMA at concentration of 3 μM. However, pretreatment with rutin reversed TEGDMA-inhibited the expression of Nrf-2 and HO-1. In addition, pretreatment with rutin enhanced the expression of Nrf-2 and HO-1 on TEGDMA-treated RAW264.7 macrophages in a concentration-dependent manner, with the upregulation becoming significant at 30 μM (*p* < 0.05).

### 2.8. Effects of Rutin on AMPK Phosphorylation Reduced by Triethylene Glycol Dimethacrylate (TEGDMA)

As shown in Figure 8, 3 μM TEGDMA reduced AMPK phosphorylation. However, pretreatment with rutin reversed TEGDMA-inhibited the phosphorylation of AMPK. In addition, pretreatment with rutin enhanced the phosphorylation of AMPK on TEGDMA-treated RAW264.7 macrophages in a concentration-dependent manner, with the upregulation becoming significant at 30 μM (*p* < 0.05).

## 3. Discussion

Macrophages play a crucial role in the immune system response against toxins and invasive pathogens, such as viruses, microbes, and toxics [24,25,26]. Overactivation of macrophages in the human body leads to the production of proinflammatory and lethal mediators, including cytokines, proteases, free radicals, and ROS [27]. Through macrophage cytotoxicity and peripheral tissue damage, these mediators cause serious and sometimes even fatal diseases, such as sepsis, infection, neurodegenerative diseases, and cancer [27]. As a resin monomer, TEGDMA is widely used in restorative resins and binding reagents after light-curing polymerization [10,11,28]. However, TEGDMA is released form polymeric materials, and it induces cytotoxicity in macrophages, pulp cells, odontoblast-like cells, and gingival fibroblasts [12,19,20,29,30,31]. Rutin is a flavonol glycoside that has several biological uses, for example, in anti-inflammation, antioxidation, anticancer, and antigenotoxicity agents [3,4,5,6]. Rutin inhibits the level of cytotoxicity in LPS- and BisGMA-treated macrophages [7,8]. In the present study, our primary goal was to determine whether the cytotoxicity induced by TEGDMA in RAW264.7 macrophages can be inhibited by rutin in a concentration-dependent manner.

Apoptosis, also called programmed cell death, plays a pivotal role in cytotoxicity. Previous studies have highlighted that TEGDMA induces apoptosis and necrosis in macrophages, preodontoblasts, and pulp cells [19,30,32]. During caecal ligation and puncture surgery, rutin inhibits cardiac apoptosis in high-glucose and sepsis-induced cardiomyopathy [33,34]. Rutin also decreases the rate of apoptosis in the liver and kidneys in rats with deltamethrin-induced hepatotoxicity and nephrotoxicity [35]. After macrophages are incubated with BisGMA, rutin decreases the rate of apoptosis and necrosis [8]. In this study, pretreatment with rutin markedly inhibited the apoptotic and necrotic effect induced by TEGDMA in macrophages. These results indicate that rutin decreases the level of TEGDMA-induced cytotoxicity by decreasing the rate of apoptosis and necrosis.

Genotoxicity, which has a damaging effect on DNA integrity, triggers apoptosis in macrophages [36,37]. In addition, TEGDMA induces DNA damage in gingival fibroblasts, lymphocytes, and oral keratinocytes [38,39,40]. The results obtained from the MN and comet assays highlight that TEGDMA induces MN formation and DNA damage parameters, including tail length and tail moment [19]. In neuron cultures of serum/glucose-deprived rats treated with 2,5-hexanedione, rutin decreased DNA fragmentation [41,42]. Rutin also inhibits genotoxicity in macrophages incubated with BisGMA [8]. Rutin considerably decreases TEGDMA-induced apoptosis by downregulating genotoxicity.

When RAW264.7 macrophages are incubated with TEGDMA, a caspase-dependent genotoxicity pathway induces apoptosis [19]. Caspases are proteolytic enzymes that play a critical role in apoptosis through genotoxicity [43]. Caspase-3 is an executioner caspase that promotes DNA strand breaking and fragmentation. The upstream factors of caspase-3 are initiator caspases, which include two categories: caspase-9 (an intrinsic pathway) and caspase-8 (an extrinsic pathway) [43]. Caspase-9 is excited directly by mitochondrial disruption or by caspase-8 cleavage through death receptor activation [43]. Rutin decreases the expression of caspase-3 in the cardiac tissue of diclofenac-treated rats, in the liver and kidney tissue of sodium-valproate-treated rats, and in cochlear hair cells after cisplatin treatment [3,44,45]. Rutin also inhibits BisGMA-induced caspase-3 and caspase-9 activation in macrophages [8]. In this study, we demonstrated that pretreatment with rutin decreased the activity of caspase-3, caspase-8, and caspase-9 in TEGDMA-treated RAW264.7 macrophages. These findings suggest that rutin can reduce TEGDMA-induced apoptosis and DNA damage through the activation of caspase-3, caspase-8, and caspase-9.

During apoptosis and genotoxicity, the accumulation of intracellular ROS plays a critical role in caspase activation [46,47]. In life-threatening situations, ROS are generated by macrophages through various processes in response to invasive pathogens [48]. However, excess production of ROS induces oxidative stress, which damages biomolecules such as DNA, lipids, and proteins. Oxidative stress is also associated with cytotoxicity through genotoxicity and apoptosis [25]. For example, TEGDMA induces oxidative stress, which in turn results in toxic effects on odontoblast-like cells, gingival fibroblasts, embryonic palatal mesenchymal cells, dental pulp cells, keratinocytes, and macrophages [21,31,40,49,50]. The data obtained in this study indicate that rutin inhibits TEGDMA-induced ROS generation in a concentration-dependent manner.

Intracellular accumulation of ROS is regulated by antioxidant system, including AOEs and antioxidant protein such as HO-1 [51,52]. Generally, AOEs, such as SOD and CAT, degrade intracellular ROS. The superoxide anion scavenger SOD converts superoxide anions into hydrogen peroxide. Subsequently, CAT catalyzes and breaks down hydrogen peroxide into water and oxygen [51]. HO-1 is a stress induced protein and plays an important role in antioxidant protection and the anti-inflammation in macrophages [50]. Expression of HO-1 and AOEs is mediated by Nrf-2, the eukaryotic redox-active transcription factor [53]. Phosphorylation of AMPK triggers Nrf-2 activation results in transactivation of downstream genes [54]. In odontoblast-like cells, TEGDMA decreases the activity of CAT [31]. Moreover, rutin decreases the level of oxidative stress by ameliorating the activity of SOD and CAT in cadmium-, valproate-, and deltamethrin-induced hepatotoxicity and nephrotoxicity [3,35,55]. We observed similar findings with rutin ameliorating the TEGDMA-inhibited activity of SOD and CAT. In addition, rutin enhances AOE activity on TEGDMA-treated macrophages. HO-1 is a stress induced protein and play the important role in antioxidant protection and the anti-inflammation in macrophages [52]. Expression of HO-1 and AOEs is mediated by Nrf-2, the eukaryotic redox-active transcription factor [53]. Phosphorylation of AMPK triggers Nrf-2 activation results in transactivation of downstream genes [54]. We first found rutin revising the HO-1 expression, Nrf2 expression, and AMPK phosphorylation slightly reduced by TEGMDA. In addition, rutin enhanced HO-1 expression, Nrf2 expression, and AMPK phosphorylation on TEGDMA-treated macrophages. These findings suggest that rutin decreases TEGDMA-induced caspase-dependent apoptosis by downregulating the generation of ROS via reactivation of antioxidant system and relative upstream factors, including Nrf-2 expression and AMPK phosphorylation.

## 4. Materials and Methods

### 4.1. Cell Culture and Treatment

The mouse macrophage cell line RAW264.7 was purchased from the cell bank of the Bioresource Collection and Research Center (Hsinchu, Taiwan). The cells were cultured in Dulbecco’s modified Eagle’s medium supplemented with 10% fetal bovine serum, 1 mM sodium pyruvate, and 1% penicillin/streptomycin/Fungizone mixture (Gibco BRL, Life Technologies, Grand Island, NY, USA) in a humidified incubator at 37 °C in the presence of 5% CO_2_ [18]. Rutin (94% pure, St. Louis, MO, USA) and TEGDMA (Sigma-Aldrich, St. Louis, MO, USA) was dissolved in dimethyl sulfoxide (DMSO; Sigma-Aldrich, St. Louis, MO, USA). The final content of DMSO in culture medium was less than 0.1% (*v*/*v*). After seeding overnight, RAW264.7 cells were plated in 24-well plate with 2 × 10^5^ cells per well. The cells were incubated with rutin concentrations of 0, 10, 30, and 100 μM for 30 min before being treated with TEGDMA of 0 or 3 μM for 24 h. The cells were incubated with rutin at 0 μM for 30 min before being treated with TEGDMA of 0 μM for 24 h. This is the negative control group, also called the control group. The cells were incubated with rutin at 0 μM for 30 min before being treated with TEGDMA of 3 μM for 24 h. This is the positive control group, also called TEGDMA group.

### 4.2. Cell Viability Assay

An MTT assay was used to determine the effect of rutin on the viability of TEGDMA-treated RAW264.7 cells. The procedure was performed according to a previous study [56]. After cell treatment, the cells were incubated with the 3-(4,5-dimethylthiazol-2-yl)-2,5-diphenyltetrazolium bromide (MTT; Sigma-Aldrich, St. Louis, MO, USA) at 5 mg/mL in phosphate buffered saline (PBS) for 4 h in the dark. Subsequently, the medium was discarded, DMSO was added, and the crystal was obtained. An absorbance value of 570 nm was determined using a microplate reader (Synergy HT, BioTek, Winooski, VT, USA).

### 4.3. Evaluation of Apoptosis and Necrosis

After TEGDMA treatment and cell trypsinisation and collection, apoptosis and necrosis were assessed using an Annexin V-FITC and propidium iodide (PI) apoptosis detection kit (BioVision, Milpitas, CA, USA). The procedure was performed according to previous studies, and flow cytometry was used for analysis of the results [57]. The results were analyzed by BD Accuri C6 flow cytometer (BD Biosciences, San Jose, CA, USA).

### 4.4. Evaluation of Genotoxicity

Genotoxicity, also called DNA damage, was analyzed using an alkaline single-cell gel electrophoresis (comet) assay and a cytokinesis-block micronucleus (MN) assay. The procedures were performed according to previous studies [25]. The comet assay results revealed that the cells were mixed with the lysis solution on microscope slides coated with low-melting-point agarose. After electrophoresis, the slides were neutralized using a neutralization buffer and stained with ethidium bromide (Sigma-Aldrich, St. Louis, MO, USA). To quantify DNA damage, the tail moment and length were evaluated using the Comet v. 3 (Kinetic Imaging Ltd., Liverpool, UK). After the cells were incubated with cytochalasin B, an MN assay was used to assess rutin and TEGDMA pretreatment. Subsequently, after the cells were washed, they were resuspended in 75 mM KCl, fixed in a 3:1 mixture of methanol/acetic acid (Sigma-Aldrich, St. Louis, MO, USA), and stained with a 3% Giemsa solution. The MN assay was performed using a light microscope.

### 4.5. Detection of Caspase Activity

Fluorometric assay kits (Enzo Life Sciences, Plymouth, PA, USA) were used to assess the activity of caspase-3, caspase-8, and caspase-9. After treatment, RAW264.7 cells were collected and lysed, and equal volumes of protein were incubated with fluorogenic substrates of caspase-3, caspase-8, and caspase-9. A fluorescence microplate reader (excitation: 485 nm, emission: 505 nm; BioTek Instruments, Winooski, VT, USA) was used to measure the relative fluorescence units (RFU) of the released 7-amino-4-(trifluoromethyl)coumarin, a fluorescent reporter molecule [25].

### 4.6. Assessment of Intracellular Reactive Oxygen Species (ROS) Generation

After rutin and TEGDMA treatment, the cells were incubated with dichlorodihydrofluorescein diacetate for 30 min. Next, the level of intracellular ROS generation was evaluated using a dichlorodihydrofluorescein diacetate (DCFH-DA) assay, as described previously [24]. Finally, a fluorescence microplate reader (excitation: 485 nm, emission: 530 nm) was used to detect fluorescence.

### 4.7. Detection of Antioxidant Enzyme (AOE) Activity

After rutin and TEGDMA treatment, the activity of AOEs, including superoxide dismutase (SOD) and catalase (CAT), was evaluated using activity assay kits (Cayman Chemical, Ann Arbor, MI, USA), according to the manufacturers’ protocols and previous studies [24].

### 4.8. Western Blot Analysis

After rutin and TEGDMA treatment, the cells were lysed by radioimmunoprecipitation (RIPA) lysis buffer containing protease inhibitors and phosphatase inhibitors as in previous studies [57]. Cellular proteins were separated using sodium dodecyl sulfate polyacrylamide gel electrophoresis and transferred to polyvinylidene fluoride membranes. The membranes were blocked and then blotted with primary antibodies against HO-1, Nrf-2, phosphorylation of AMPK, AMPK, β-actin (Santa Cruz Biotech, Dallas, TX, USA) overnight at 4 °C. The membranes were washed and further incubated with secondary antibodies. Finally, the membranes were imaged by enhanced chemiluminescence kit using the Fusion Solo S (Vilber Lourmat, Collégien, France). The band intensities of protein expression and phosphorylation were quantified using Evolution Capt software (Vilber Lourmat fusion solo S). The levels of protein expression and phosphorylation were expressed relative to endogenous control levels.

### 4.9. Statistical Analysis

The data presented in all figures were analyzed using SPSS and are expressed as mean ± standard deviation (S.D.). All statistical analyses were performed using one-way analysis of variance (ANOVA) followed by a Bonferroni multigroup comparison test. A *p*-value less than 0.05 was considered significant for all tests.

## 5. Conclusions

In summary, we demonstrated that rutin can prevent TEGDMA-induced cytotoxicity in RAW264.7 macrophages, mainly through the downregulation of apoptosis. Moreover, the inhibition of apoptosis is invariably associated with a reduction in genotoxicity in RAW264.7 macrophages incubated with TEGDMA along with rutin pretreatment. We further discovered that rutin inhibited the activity of caspase-3, caspase-8, and caspase-9, which mediated apoptosis through genotoxicity in TEGDMA-treated RAW264.7 macrophages. Rutin also inhibited the activity of caspase-3, caspase-8, and caspase-9 by downregulating the generation of ROS. Rutin-induced downregulation of ROS generation is mediated by upregulation of AOE activity and HO-1 expression via Nrf-2 expression and AMPK phosphorylation. These findings highlight the novel protective mechanism of rutin against TEGDMA-mediated toxic effects in RAW264.7 macrophages.

## Figures and Tables

**Figure 1 ijms-23-11773-f001:**
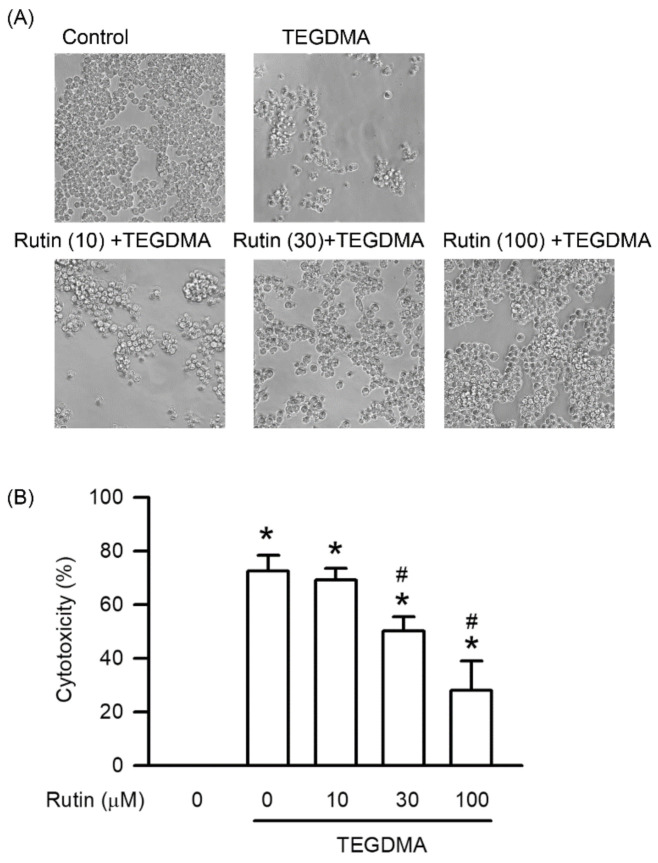
Effect of rutin on TEGDMA-induced cytotoxicity in RAW264.7 macrophages. After RAW264.7 macrophages were incubated with rutin at 0, 10, 30, and 100 μM for 30 min, the cells were incubated with or without TEGDMA at 3 μM for 24 h. (**A**) The image of morphological changes was taken using the inverted microscope at the actual magnification 200×. (**B**) Cytotoxicity was represented as the percentage of the control group, which indicated 0% cytotoxicity rate in the treatment with TEGDMA and rutin at 0 μM. Values are expressed as mean ± S.D. of three times per group. * Represents significant difference between the indicated and control groups; # between the indicated and TEGDMA groups, *p* < 0.05.

**Figure 2 ijms-23-11773-f002:**
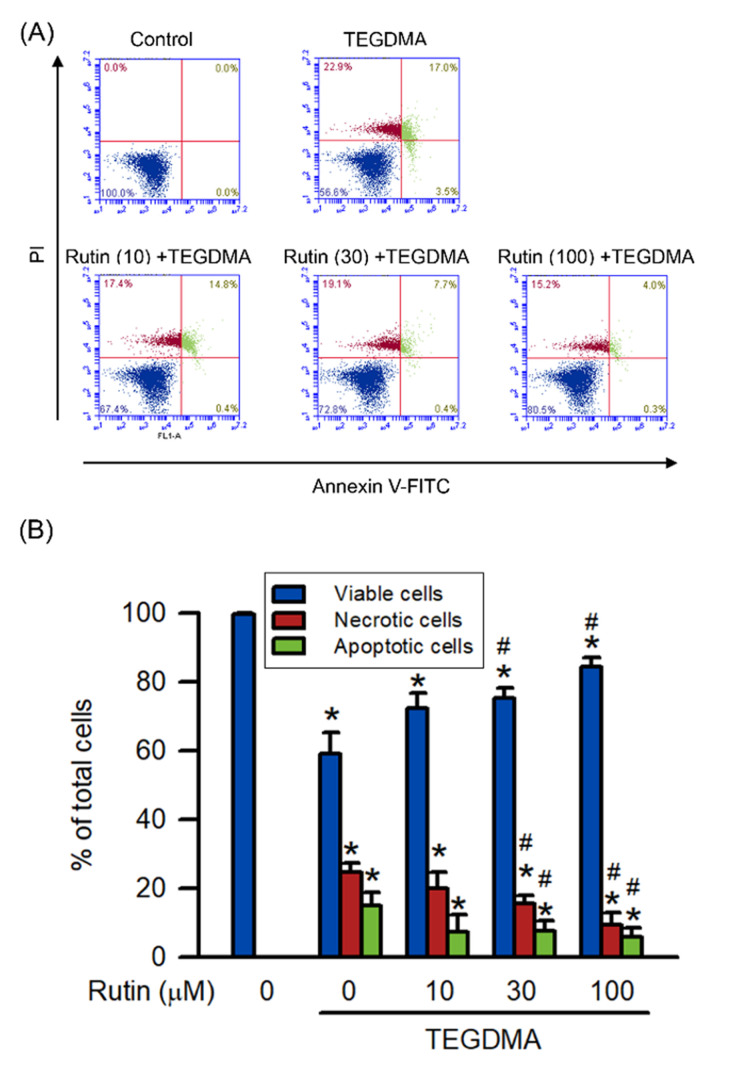
Effect of rutin on TEGDMA-induced necrosis and apoptosis expression in RAW264.7 macrophages. Apoptosis of RAW264.7 cells was measured by flow cytometry using Annexin V-FITC/PI apoptosis detection kit. (**A**) The dot-plots of flowcytometry from the annexin V-FITC/PI staining assay are shown. (**B**) Quantitative analysis of the percentage of viable cells (Annexin V-FITC negative and PI negative), necrotic cells (Annexin V-FITC negative and PI positive), and apoptotic cells (Annexin V-FITC positive) was presented. Values are expressed as mean ± S.D. of three times per group. * Represents significant difference between the indicated and control groups, *p* < 0.05. # Means significant difference between the indicated and TEGDMA groups, *p* < 0.05.

**Figure 3 ijms-23-11773-f003:**
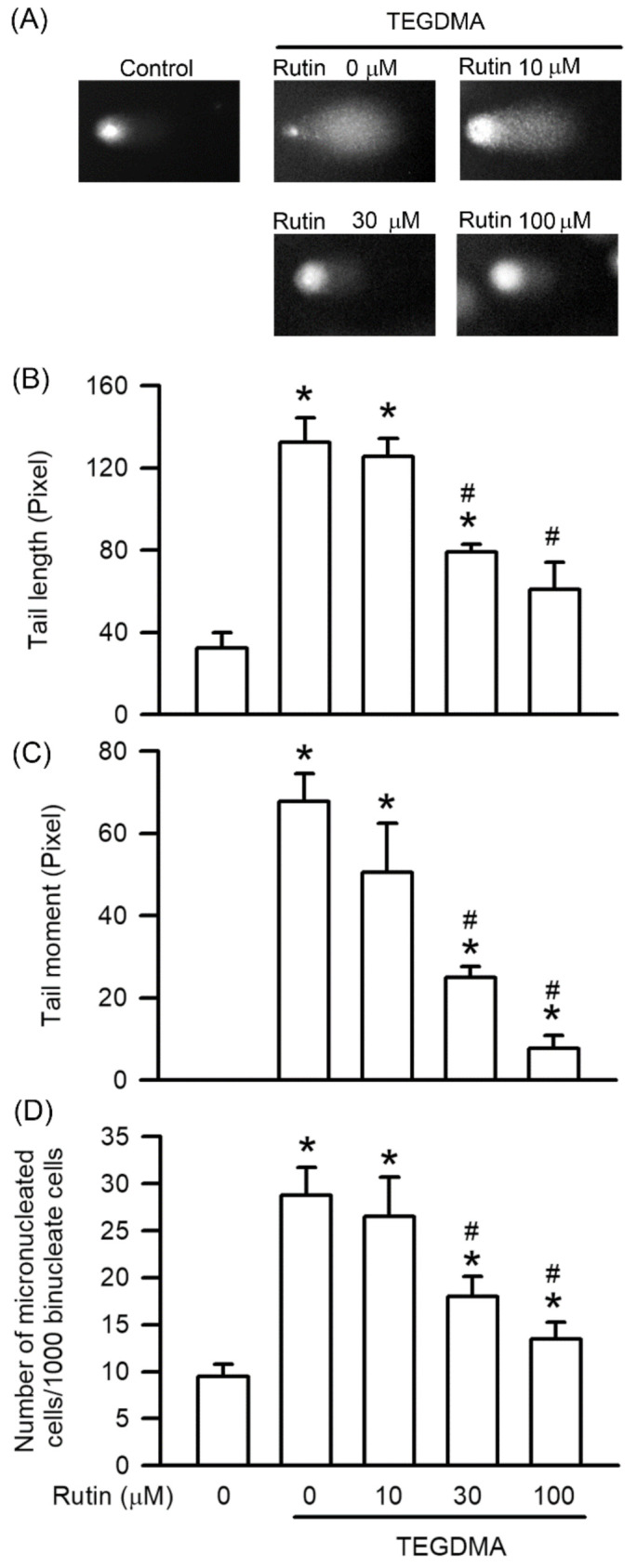
Effect of rutin on TEGDMA-induced genotoxicity via COMET and MN assay in RAW264.7 macrophages. (**A**) was gel electrophoresis of DNA damage from COMET assay. (**B**,**C**) were quantifications of tail length and tail moment, respectively. (**D**) was the quantifications of MN formation. Values are expressed as mean ± S.D. of three times per group. * Represents significant difference between the indicated and control groups, *p* < 0.05. # Means significant difference between the indicated and TEGDMA groups, *p* < 0.05.

**Figure 4 ijms-23-11773-f004:**
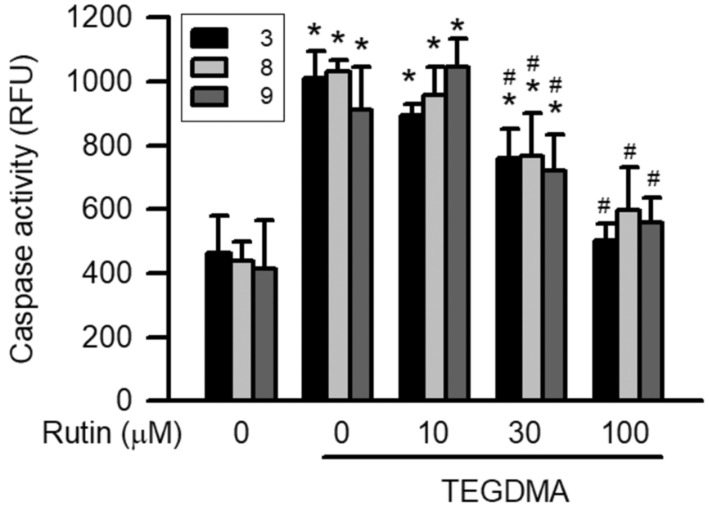
Effect of rutin on TEGDMA-induced caspases activities in RAW264.7 macrophages. Values are expressed as mean ± S.D. of three times per group. * Represents significant difference between the indicated and control groups, *p* < 0.05. # Means significant difference between the indicated and TEGDMA groups, *p* < 0.05.

**Figure 5 ijms-23-11773-f005:**
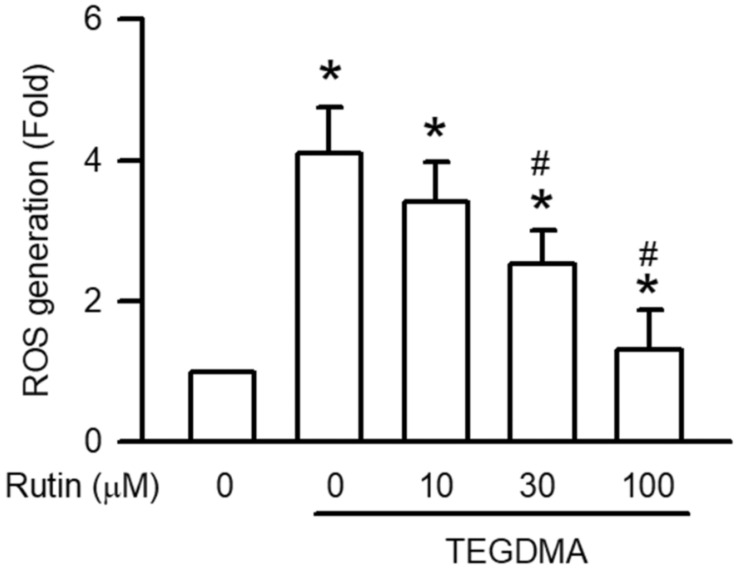
Effect of rutin on TEGDMA-induced ROS generation in RAW264.7 macrophages. Values are expressed as mean ± S.D. of three times per group. * Represents significant difference between the indicated and control groups, *p* < 0.05. # Means significant difference between the indicated and TEGDMA groups, *p* < 0.05.

**Figure 6 ijms-23-11773-f006:**
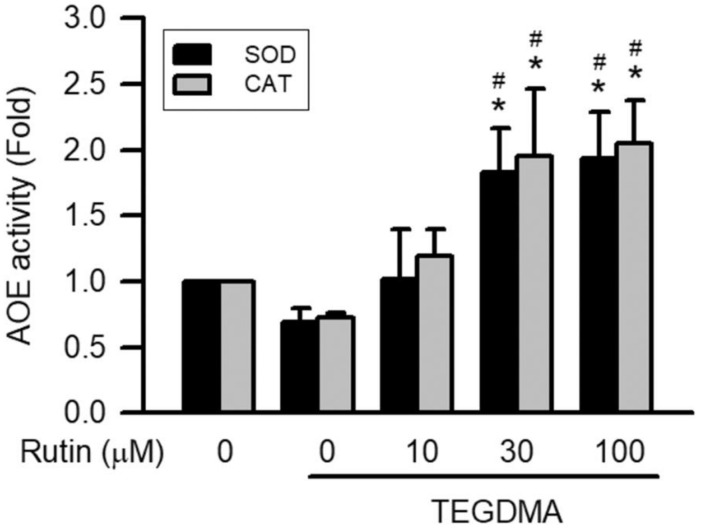
Effect of rutin on TEGDMA-reduced AOE activities in RAW264.7 macrophages. Values are expressed as mean ± S.D. of three times per group. * Represents significant difference between the indicated and control groups, *p* < 0.05. # Means significant difference between the indicated and TEGDMA groups, *p* < 0.05.

**Figure 7 ijms-23-11773-f007:**
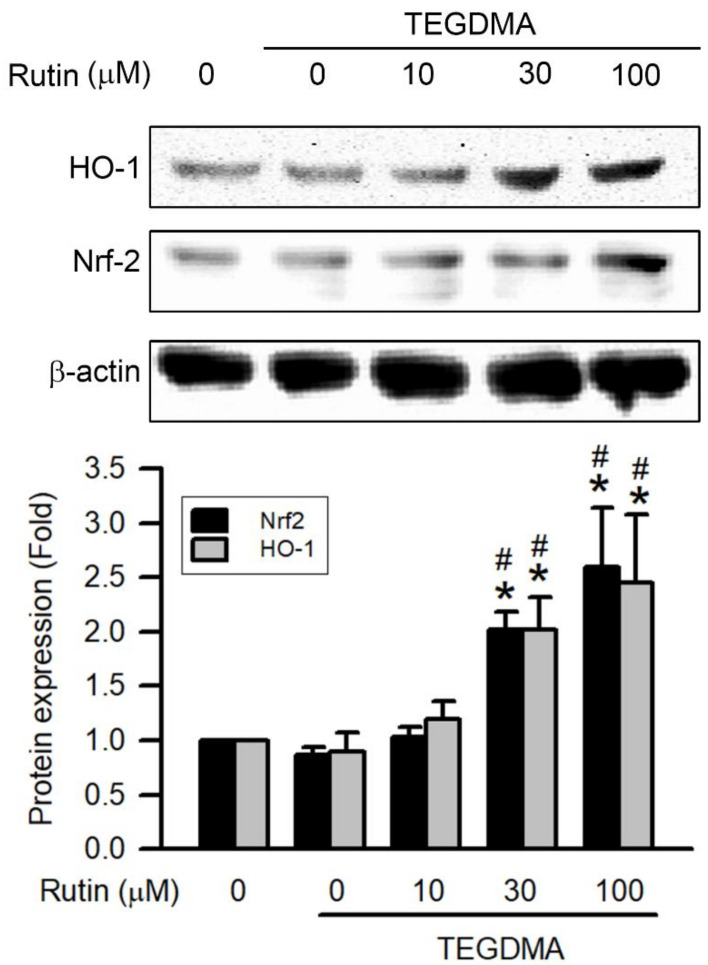
Effect of rutin on HO-1 and Nrf-2 expression in TEGDMA-treated RAW264.7 macrophages. Expression of HO-1 and Nrf-2 was determined using Western blotting with indicated antibodies. Representative blots and quantitative results are shown. Values are expressed as mean ± S.D. of three times per group. * Represents significant difference between the indicated and control groups, *p* < 0.05. # Means significant difference between the indicated and TEGDMA groups, *p* < 0.05.

**Figure 8 ijms-23-11773-f008:**
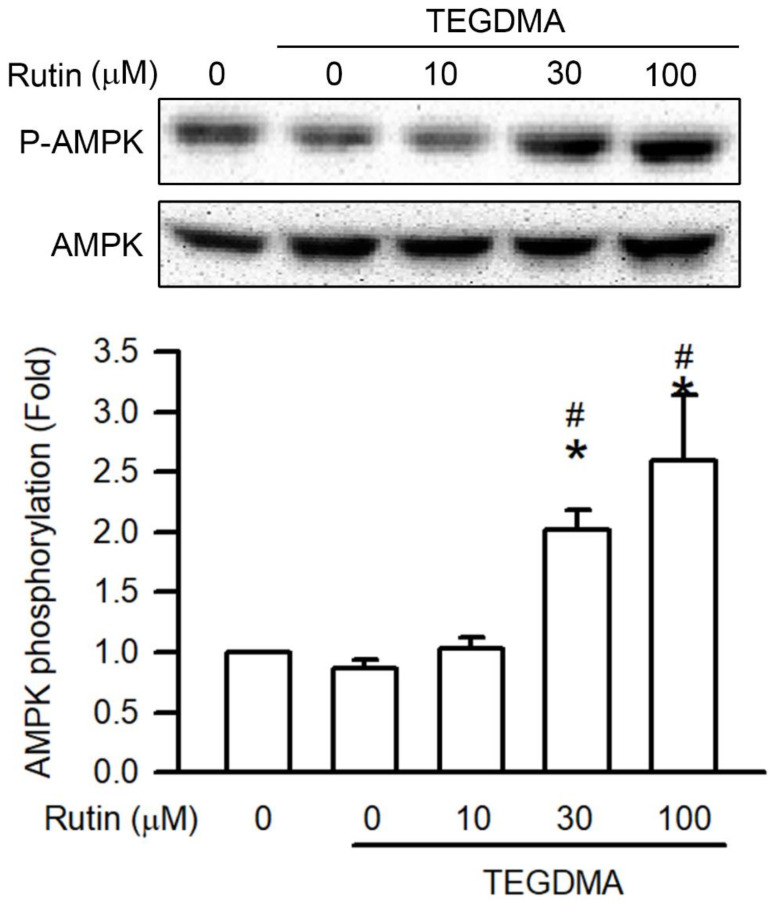
Effect of rutin on AMPK phosphorylation in TEGDMA-treated RAW264.7 macrophages. Phosphorylation of AMPK was determined using Western blotting with indicated antibodies. Representative blots and quantitative results are shown. Values are expressed as mean ± S.D. of three times per group. * Represents significant difference between the indicated and control groups, *p* < 0.05. # Means significant difference between the indicated and TEGDMA groups, *p* < 0.05.

## Data Availability

Not applicable.

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
