# Peer review of "Protective Effect of Rutin on Triethylene Glycol Dimethacrylate-Induced Toxicity through the Inhibition of Caspase Activation and Reactive Oxygen Species Generation in Macrophages"

_ijms, 2022, doi:10.3390/ijms231911773_

Round 1

Reviewer 1 Report (Previous Reviewer 1)

The authors have done all the required revisions and the quality of the manuscript was improved.

Author Response

Thank you for your recognition of our work.

Reviewer 2 Report (New Reviewer)

1. Images of cells at the stage when TEGDMA was added and 24 hours after exposure should be presented. I propose to combine with figure 1.

2. The quality of Figure 2 needs to be improved. It needs to be done in color. Applies to the rest of the illustrations.

3. The method of counting (analyzing) Western blots should be described in detail.

4. According to the rules of the journal, the material and methods should be placed after the discussion.

5. A separate conclusions section must be submitted.

6. The article should be seriously proofread and references checked. For example, Recent several studies have purposed that cytotoxicity induced by TEGDMA 64 in RAW264 macrophages via their large interaction potency and impregnation into lipid bilayers [Murakami et al., 2019]. The numbering of links should be continuous in the form of numbers.

7. In the introduction, the authors should briefly characterize the functions of macrophages and the significance of ROS generation in their functioning. The importance and relevance of the work done should be clearly indicated.

Author Response

We thank deeply to all the constructive and instructive comments from the reviewer. We had considered each comment seriously and appropriate revision was made according to the editor’s and reviewers’ suggestions.

1. Images of cells at the stage when TEGDMA was added and 24 hours after exposure should be presented. I propose to combine with figure 1.

Response:

We deeply thank the reviewer for the suggestion. We have added the image of morphological changes in Figure 1A.

2. The quality of Figure 2 needs to be improved. It needs to be done in color. Applies to the rest of the illustrations.

Response:

We deeply thank the reviewer for the suggestion. We have improved the quality of figure 2. In addition, we have added the color in Figure 2.

3. The method of counting (analyzing) Western blots should be described in detail.

Response:

We deeply thank the reviewer for the friendly reminder. We have described the quantification of western blot in detail in the section of 4.8 Western blot analysis.

4. According to the rules of the journal, the material and methods should be placed after the discussion.

Response:

We deeply thank the reviewer for the friendly reminder. We have placed the material and methods after the paragraph of discussion.

5. A separate conclusions section must be submitted.

Response:

We deeply thank the reviewer for the friendly reminder. We have separated the paragraph of the conclusion.

6. The article should be seriously proofread and references checked. For example, Recent several studies have purposed that cytotoxicity induced by TEGDMA 64 in RAW264 macrophages via their large interaction potency and impregnation into lipid bilayers [Murakami et al., 2019]. The numbering of links should be continuous in the form of numbers.

Response:

We deeply thank the reviewer for the friendly reminder. We have rechecked the numbers of references seriously.

7. In the introduction, the authors should briefly characterize the functions of macrophages and the significance of ROS generation in their functioning. The importance and relevance of the work done should be clearly indicated.

Response:

We deeply thank the reviewer for the suggestion. We have added the sentences in the paragraph of introduction.

Round 2

Reviewer 2 Report (New Reviewer)

The article is ready for publication. I recommend that you take it as presented.

Author Response

Thank you for your recognition of our work.

This manuscript is a resubmission of an earlier submission. The following is a list of the peer review reports and author responses from that submission.

Round 1

Reviewer 1 Report

This work is considered of interest.  In my opinion, the manuscript is suitable for publication in the International Journal of Molecular Science, after the authors have addressed the following comments and questions. Overall, a little bit of attention is considered to the construct and use of grammar, scientific writing, and in-text referencing style.

·       Line 5: Is “and PhD” a part of the last author's name?

·       Lines 41-42: The sentence is difficult to understand, please rewrite the sentence. The use of the verb “present” is inappropriate.

·       Lines 42-43: The sentence needs a revision to be more comprehensive. The use “abundantly available” is not suitable. It is better to use “found in abundance” or “abundantly found”.

·       Line 46: the word “pre-treatment” was written in a different way than the whole text.

·       The authors should mention the source for rutin and its purity in the material section.

·       The authors should mention the solvents in which rutin and TEGDMA were prepared.

·       The authors should add more information about the used control (negative and positive) groups to the material and method section.

·     Line 139: It should be better to add the abbreviation (TEGDMA) to the header beside its full name.

·       Figure 1 (line 145-146) # “means significant” should be added.

·    Figure 2: A and B letters used in the figure should be mentioned in the figure capture.

·       Figure 3: sections C and D are missed, and B is wrongly placed.

·       Lines 231-233: Please add references in the sentence.

·       Please check the grammar mistakes all over the manuscript for example the verbs in lines 297, and 298.

·       Line 302: The sentence is difficult to understand, please rewrite the sentence to be more comprehensive.

References need more revision. For example the format of references 1 and 2.

Reviewer 2 Report

In this manuscript, the authors try to demonstrate the cytoprotection effects of rutin against TEGDMA toxicity using macrophage cells. It has been reported that the toxicity of triethylene glycol dimethacrylate (TEGDMA), which is utilized as polymeric biomaterials, induces cell death through caspase-dependent and caspase independent pathways. Therefore, this manuscript can be a good example of a research to prevent the toxicity of TEGDMA. However, this manuscript cannot be accepted into this journal for the following reasons.

Major concern:

1. Abstract part, the authors mentioned that “Finally, we found that TEGDMA-induced HO-1 expression, Nrf-2 expression, and AMPK phosphorylation would be inhibited by rutin.”---- in Figures 7 and 8, there was no significant evidence that TEGDMA affects HO-1 expression, Nrf-2 expression, and AMPK phosphorylation in this data. However, rutin treatment increases HO-1 expression, Nrf-2 expression, and AMPK phosphorylation. I would suggest that the authors need to reflect the contents of maintext in the abstract.

There is no available explanation regarding cell viability and these protein expression. It may be reasonable if the authors show the mechanism of action of rutin on the apoptosis pathway in macrophage.

2. Introduction part, the significance of TEGDMA toxicity in macrophages or rationale for this research are not sufficiently presented. I would suggested that authors may need to present the significance of this research in the manuscript.  

3. In line 140 and figure 1, the authors do not provide the cytotoxicity of TEGDMA treatment. In the maintext, the author mentioned 3uM of TEGDMA was treated to raw 264.7 cell to induce toxicity. The cell viability (%) of 3uM of TEGDMA is not clearly presented. It is difficult to evaluate the protective effects of rutin without cell viability of TEGDMA treatment.

The author mentioned that TEGDMA was treated with 3uM. But they do not provide detail condition of cell viability procedure in experimental part or maintext. For example, incubation time, cell seeding number, plate type, DMSO amount, MTT kit number and etc. The authors mention reference 17 is used for cell viability assay. But reference 17 is a toxic study of BrFlu using zebrafish, so that the reference 17 is not related with this experiment. In the data of figure 1, it is unclear whether figure 1 shows the protective effects of rutin against TEGDMA or the toxicity of rutin.

4. figure 2, A), B), C) No detailed explanation of the legend. The authors need to mention in the main text and legend of figure 1. A) facs data does not have labels. so it is unclear what experiment they performed. It seems that the experiments were performed using annexin v vs PI staining, which can distinguish ratio of live and death cells. In this data, the ratio of live cell in TEGDMA treatment group was 71.6%. The ratio of live cells in the rutin and TEGDMA cotreatment group was 75%, and the only rutin 30 was 87.7%. Considering the FACS data, the concentration range of the protection effect for rutin is narrow, so that the protection effect can be very limited. In case of B), the ratio of live cells increases as the dose dependently. However, it is inconsistent with the results of experiment A). It seems that the results of rutin 100 and rutin 30 were written interchangeably.

5. In the experiment of figure 5, the caspase activity seems to decrease as the concentration of rutin increases. There is no correlation with the FACS results. The author need to discuss in the maintext. Again, figure 1 data does not exactly present the protective effect of rutin against TEGDMA treatment. FACS data is the result for cytoprotective effect of rutin.

6. In figure 7 and figure 8, it is necessary to explain how the activity of HO-1 and NRF2 are affected by TEGDMA treatment. How is it related to apoptosis? Western blot data provided by the author showed no effect on HO-1 and NRF2 expression on TEGDMA treatment. The expression levels of HO-1 and NRF2 are dependent on concentration of rutin. The expression level of AMPK and p-AMPK are dependent on the concentration of rutin too.

7. The authors need to improve the detail description of manuscript including introduction, experimental section and maintext for potential readers.